# Rheological Behaviour of Highly Filled Materials for Injection Moulding and Additive Manufacturing: Effect of Particle Material and Loading

**Marko Bek** [1] **, Joamin Gonzalez-Gutierrez** [2] **, Christian Kukla** [3] **,**
**Klementina Pušnik Črešnar** [4] **, Boris Maroh** [5] **and Lidija Slemenik Perše** [1,*]

1   Faculty of Mechanical Engineering, University of Ljubljana, 1000 Ljubljana, Slovenia; marko.bek@fs.uni-lj.si
2   Institute of Polymer Processing, Montanuniversitaet Leoben, 8700 Leoben, Austria;
    joamin.gonzalez-gutierrez@unileoben.ac.at
3   Industrial Liaison, Montanuniversitaet Leoben, 8700 Leoben, Austria; christian.kukla@unileoben.ac.at
4   Faculty of Mechanical Engineering, University of Maribor, 2000 Maribor, Slovenia; klementina.pusnik@um.si
5   Polymer Competence Center Leoben GmbH, 8700 Leoben, Austria; boris.maroh@pccl.at
*   Correspondence: lidija.slemenik.perse@fs.uni-lj.si; Tel.: +386-6207-100

**Abstract:** Within this paper, we are dealing with a mixture of thermoplastic polymer that is filled with inorganic fillers at high concentrations up to 60 vol.%. A high number of particles in the compound can substantially change the rheological behaviour of the composite and can lead to problems during processing in the molten state. The rheological behaviour of highly filled materials is complex and influenced by many interrelated factors. In the present investigation, we considered four different spherical materials: steel, aluminium alloy, titanium alloy and glass. Particles with similar particle size distribution were mixed with a binder system at different filling grades (30–60 vol.%). We showed that the rheological behaviour of highly filled materials is significantly dependent on the chemical interactions between the filler and matrix material. Moreover, it was shown that the changes of the particle shape and size during processing lead to unexpected rheological behaviour of composite materials as it was observed in the composites filled with glass beads that broke at high contents during processing.

**Keywords:** highly filled materials; polypropylene binder; binder-filler interaction; particle–particle interaction; shear rheology; fused filament fabrication; ATR-FTIR; XPS

## 1. Introduction

Highly filled polymers are polymer composites with particles, added at concentrations well above 20 vol.%. Due to the high concentration of the particles, the interactions between the particles cannot be neglected [1]. Highly filled polymers include fibre-reinforced polymers, wood–polymer composites [2–4], bonded magnets [5–7], and feedstock materials for injection moulding and additive manufacturing [8–12]. They are used in automotive, aerospace, electronics, construction, and consumer goods industries.

The focus of the present investigation is on highly filled polymers with up to 60 vol.% of sinterable particles. One of the major applications of highly filled polymers with more than 40 vol.% is the indirect fabrication of metal, ceramic, or cermet parts with complex geometry in processes like powder injection moulding (PIM) [13], powder extrusion moulding (PEM) [14,15], and more recently, powder fused filament fabrication (PFFF) [10]. All these technologies follow three main steps after the preparation of the highly filled thermoplastic compound. First, the part is shaped, then the polymeric fraction of the material is removed, and finally, the part is sintered in a furnace. These processes are usually

named as Shape, Debind, and Sinter (SDS) processes [10]. Feedstock materials used in SDS processes have a complex multimaterial formulation since they meet several specific requirements for each of the stages of the process. The binder system usually consists of the main binder component, a backbone, and additives. The main binder component usually provides flowability and is removed first during debinding. The backbone supplies shape retention during debinding and is removed last. Additives help to improve various properties of the particle dispersion in the main binder component or prevent binder degradation.

In the SDS processes, it is always beneficial to have the highest amount of filler particles to minimise geometrical distortions and shrinkage during sintering [16]. However, a high number of particles in the compound can lead to several problems during processing in the molten state. Moreover, it is challenging to obtain a homogeneous dispersion of the particles in the polymeric binder since some voids in the feedstock could appear due to deficiency of the polymer to cover each of the individual particles [16,17].

The success of the shaping step of the SDS process depends on the rheological properties of the feedstock material. The introduction of fillers into the polymer, especially at high concentrations, can have a substantial effect on rheological properties. Several review papers are describing the behaviour of highly filled materials, such as review paper prepared by White [18], a review oriented on experimental properties of filled thermoplastic melts by Khan and Prud'Homme [19], and more recently, a review focusing on powder injection moulding and rheological properties of feedstock [20] and an extensive review of highly filled polymers by Rueda et al. [21]. The rheology of highly filled polymers is influenced by many interrelated factors such as polymeric binder and its composition, particle loading, particle shape, and size distribution and surface chemistry of the particles.

In general, higher powder loadings lead to higher viscosities and dynamic rheological moduli (i.e., G' and G") [1,16,17,22,23]. At a certain particle loading level, a particle–particle network is established that promotes highly non-Newtonian behaviour. At low shear rates, the network prevents the flow of material; however, at the specific limit, the particle network breaks and material starts to flow (apparent yield behaviour). This flow is highly affected by the particles [24,25]. The network also dictates the dynamic behaviour of materials as it limits the frequency dependence of dynamic properties [24] and at small loads result in solid-like material behaviour [25].

The rheological properties of highly filled materials depend on the selected binder and its composition as well. In our earlier work [1] we have investigated the viscosity of highly filled materials using different binder compositions and powder loadings. We have shown that the binder composition has a significant role in the behaviour of filled materials and that the choice of the binder directly influences shear viscosity of the composite materials as the binder material with lower viscosity will result in overall lower viscosity of composite material. Similarly, other researchers have investigated different thermoplastic binders for highly filled lead zirconate titanate feedstocks [26]. Among the investigated binders, they showed that low-density polyethylene is a preferred choice for the binder since it exhibited low viscosity and low die swell. Multicomponent binder systems are frequently used in SDS processes, where the reduction of viscosity can be achieved by the addition of a low molecular weight component to the binder system (i.e., waxes and stearic acid) [27].

Particle size and size distribution is yet another essential factor influencing rheology of composite materials. Usually, bigger particles result in lower viscosities and viscoelastic moduli (G' and G"), while smaller particles have a large surface area and tend to agglomerate, which results in higher viscosities and viscoelastic moduli [28]. The influence of particle size is closely related also to powder loading. Sotomayor et al. [29], investigated the influence of powder particle size distribution and solid loading content of 316L feedstocks on rheological properties. They observed that the viscosity increased as the particle size decreased; however, this was the case only for high powder loading, while for powder loadings less than 65 vol.% the viscosity changes were less significant.

Effect of filler shape can be explained through aspect ratio, which is the deviation of geometry from an ideal spherical shape. It is known that as the aspect ratio increases, the friction between polymer

binder and the fillers increases, which hinders the movement of polymeric chains and results in higher thickness [21]. For example, different viscosity values were shown for PLA (Polylactide) mixed with copper particles with different shapes (aspect ratios) [30]. For stainless steel 17-4PH powders mixed with a wax-based binder, the researchers [31] reported higher viscosities for PIM feedstock with a higher particle surface area. In the study of Dimitri et al. [32], the authors studied the effect of shape and size of stainless steel 17-4PH powders on rheological properties of feedstock material. Their results showed that both the size and shape of the particles affected the rheological behaviour of the feedstock. However, in the case of powders, the influence of the size was more significant compared to the shape of the particles. In another study [33], the researchers investigated the rheology of highly filled Inconel feedstock alloy 718 and observed that shear viscosity decreased with increasing particle sizes at a fixed 50% filler load.

Additives that are mixed with polymer and fillers significantly affect the rheological properties of composite materials. They can act as coupling agents between the filler and polymer, wetting agents, or as an internal lubricant [19]. Surface treatment disperses and wets the fillers, which results in reduced viscosity and lower G′ values [19,34,35]. Some authors have also reported that surface-treated fillers result in reduced magnitudes of steady-state viscosity, decreased apparent yield values, and lower dynamic moduli. They hypothesised that the coupling agent tied to the filler surface and on the other end of the hydrocarbon chain enabled connectivity with the polymer [34,36]. While the need for additives, especially in highly filled polymers, is generally recognised, the exact mechanisms of how they work have not been clearly understood, yet [21].

In summary, the rheological behaviour of highly filled materials is complex, and as previously noted, several factors affecting rheological behaviour are interrelated. In the present study, we reduced these factors to the investigation of different filler materials and filler loading on dynamic rheological properties. Different spherical fillers with similar particle size distribution but different chemical composition were used. Filler materials were introduced into the binder system with a fixed composition at different filling grades (30–60 vol.%).

## 2. Materials and Methods

### 2.1. Materials

A three-component binder system was used for the study. Polypropylene-ethylene copolymer (PP), Licocene PP1602, Clariant Produkte GmbH, Frankfurt, Germany was used as the main component. The second component, which was used as a backbone component in the binder was a polypropylene functionalised with maleic acid anhydride (MAPP), Scona TPPP 9212, Byk Chemie GmbH, Germany. The last component of the binder system was thermal stabiliser (TS), Add-Vance TH130, Byk Chemie GmbH, Germany, which was used to prevent thermal degradation of the composites.

The binder system was used to prepare four highly filled composites by mixing it with inorganic spherical fillers: aluminium alloy, 316L stainless steel, titanium alloy, and hollow glass beads. For the systematic study, three filler loadings: 30, 45 and 60 vol.% were prepared. The binder composition was adjusted to the filler loading. The amount of MAPP increased from 3 vol.% of main PP for filler loadings of 30 vol.% to 6 vol.% of main PP for filler loadings of 60 vol.%. The amount of MAPP component in the binder system was increased with increasing filler concentration to provide better filler–binder compatibility and was constant in relation to the filler loading (10%). For all composites, 2 wt.% of thermal stabiliser was added to the main PP. The exact binder composition for each filler loading level is presented in Table 1.

**Table 1.** Composition of the binder system used for different filler loading levels.

| Filler Loading (vol.%) | Main PP (vol.%) | Backbone MAPP (vol.%) | Stabiliser TS (wt.% of Main PP) |
|---|---|---|---|
| 30 | 67.96 | 2.04 (3% of main PP) | |
| 45 | 52.63 | 2.37 (4.5% of main PP) | 2 |
| 60 | 37.74 | 2.26 (6% of main PP) | |

Filler Materials and Filler Size Distributions

As mentioned above, four spherical filler materials were selected for the investigation. Aluminium alloy spherical particles AlSi10Mg (denoted as Al) were obtained from IMR Metal Powder Technologies GmbH, Velden am Wörthersee, Austria; stainless steel spherical particles Micro-Melt 316L were obtained from Carpenter Powder Products AB, Torshälla, Sweden; titanium alloy spherical particles Ti6Al4V (denoted as Ti) were obtained from TLS Technik GmbH & Co. Spezialpulver KG, Bitterfeld-Wolfen, Germany, and glass beads 3M™ Glass Bubbles iM16K (denoted as GB) were obtained from 3M, Minneapolis, MN, USA.

The particle size distribution of filler materials was determined by using laser diffraction by Particle size analyser (PSA 1190), Anton Paar, Graz, Austria. A small amount of powder was fed to the water that was circulating in the analyser. Before each measurement, the particles–water mixture was exposed to ultrasound for 10 s to break up the agglomerated particles and prevent re-agglomeration. The first step was followed by 30 s of particle size measurement. During this time, the particles–water mixture passed through the analyser several times to obtain average size distributions of filler materials. The volume size distribution of filler materials is presented in Figure 1.

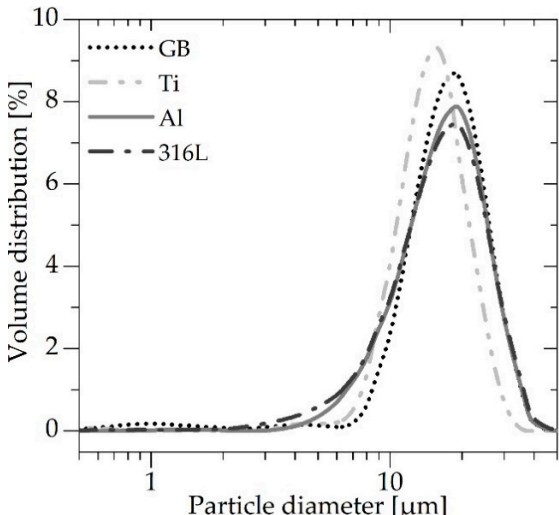

**Figure 1.** The volume size distribution of filler materials.

From Figure 1, the size parameters can be obtained for all the materials used in the study. The obtained values with the density of filler materials as provided by material suppliers are presented in Table 2.

The shape of the particles was investigated using scanning electron microscopy (SEM) JSM-IT100, JEOL, Tokyo, Japan, for which the micrographs are shown in Figure 2.

**Table 2.** Physical properties of filler materials.

| Filler Material | Size Parameters | | | | | Density (g cm⁻³) |
|---|---|---|---|---|---|---|
| | D10 (µm) | D50 (µm) | D90 (µm) | Mean (µm) | Span | |
| Aluminium alloy (Al) | 8.57 | 16.02 | 25.09 | 17.29 | 1.03 | 2.67 |
| 316L stainless steel (316L) | 7.62 | 15.65 | 25.41 | 16.94 | 1.14 | 7.99 |
| Titanium alloy (Ti) | 9.15 | 14.26 | 20.89 | 15.19 | 0.82 | 4.45 |
| Glass beads (GB) | 9.84 | 15.57 | 25.15 | 17.71 | 0.92 | 0.6 |

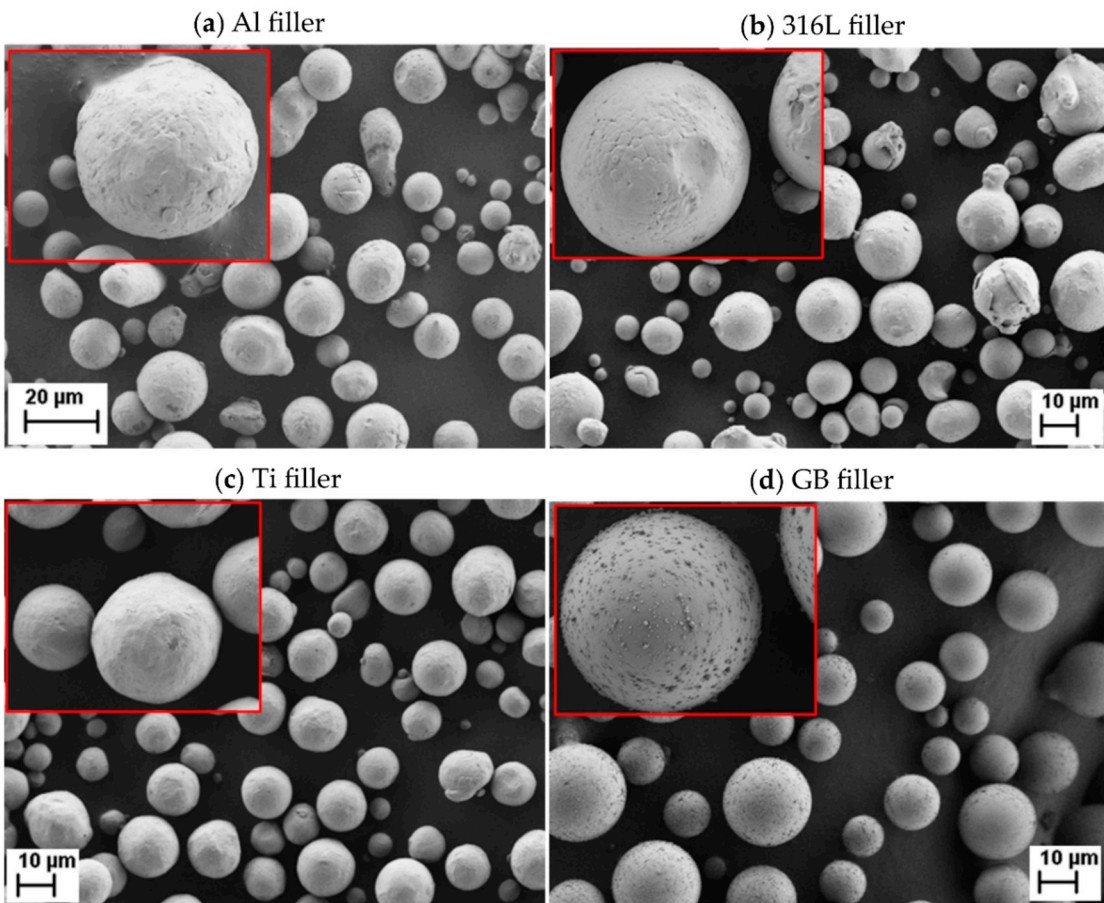

**Figure 2.** Scanning electron microscopy (SEM) micrographs of spherical filler materials: (**a**) AlSi10Mg aluminium alloy; (**b**) 316L stainless steel particles; (**c**) Ti6Al4V titanium alloy; (**d**) iM16K glass beads. The scale does not apply for the zoomed images in upper left corners.

The particle size distribution and SEM images showed that all the particles exhibited spherical shapes and comparable size distributions. Mean values of diameters for Al, 316L and GB ranged from 16.94 to 17.71 µm, while the values of the diameters for Ti particles slightly deviated with the lowest mean value of 15.19 µm. A similar trend can be observed for size parameters D10, D50, and D90. It can also be seen that all the particles exhibited comparable size distributions with span around 1. Having similar particle shape, sizes, and size distributions means that the surface area of particles was also comparable and should have a similar impact on the rheological behaviour of investigated materials.

### 2.2. Compounding and Sample Preparation

Highly filled composites were prepared by mixing the melted binder system with the fillers in Brabender kneader, Brabender Technologie GmbH, Duisburg, Germany. The kneader was fitted with counter-rotating roller rotors and a mixing chamber with a capacity of 40 cm$^3$. The mixing chamber was preheated for 30 min to 175 °C (approximately 15 °C above the melting temperature of the grafted polypropylene—MAPP), and the rotational speed was set to 60 rpm. All three binder components were introduced into the mixing chamber at the same time. Binder components were kneaded for approximately 3 min to achieve homogeneous melt. The filler powders were introduced into a homogeneously melted binder system and carefully mixed for additional 12 min.

In addition to the samples with four filler materials and three different loadings, the samples of binder systems for all loadings without a filler were prepared. Altogether, 15 different composite combinations were prepared. The complete overview of the materials and their designated names are given in Table 3.

**Table 3.** Combinations of the prepared highly filled composites and their designated names.

| Filler Material | Loading Level (Vol.%) | | |
|---|---|---|---|
| | 30 | 45 | 60 |
| Aluminium alloy (Al) | Al_30 | Al_45 | Al_60 |
| 316L stainless steel (316L) | 316L_30 | 316L_45 | 316L_60 |
| Titanium alloy (Ti) | Ti_30 | Ti_45 | Ti_45 |
| Glass beads (GB) | GB_30 | GB_45 | GB_60 |
| Binder (B) | B_30 | B_45 | B_60 |

After kneading and cooling, the mixed materials were ground in a cutting mill SM 200, Retsch GmbH, Haan, Germany, fitted with a mesh of square openings 4 × 4 mm. Granulates were collected in the mill and compression moulded to disks with diameter 25 mm and thickness of 2 mm. A vacuum press, Dr. Collin GmbH, Maitenbeth, Germany with a five-stage program was used to prepare disks by using a mould with nine round cavities. The temperature and pressure profiles used during compression moulding are shown in Table 4.

**Table 4.** Parameters used in the vacuum press during compression moulding of disk specimens.

| Stage (No.) | Temperature (°C) | Pressure (MPa) | Time (Min) |
|---|---|---|---|
| 1 | 200 | 0.1 | 20 |
| 2 | 200 | 5 | 10 |
| 3 | 200 | 6 | 10 |
| 4 | 200 | 7 | 10 |
| 5 | 30 | 7 | 20 |

### 2.3. Rheological Characterisation

Rheological characterisation of the prepared disks was performed by the rotational rheometer MCR 302 Anton Paar GmbH, Graz, Austria. For the binder systems (B) a cone–plate configuration with a diameter of 50 mm was used, while for the composites with 30 and 45 vol.% filler loading a plate–plate configuration with a diameter of 25 mm was used. For the composites with the highest loading, i.e., 60 vol.%, a serrated plate with a diameter of 25 mm was used to prevent wall slip of the composites during the measurements. The temperature for rheological tests followed the temperature of kneading (175 °C).

For rheological characterisation strain amplitude tests at a constant frequency and frequency sweep tests at the constant strain in the linear viscoelastic range were performed. Both tests were

performed as shear deformation-controlled tests, where the samples were loaded with a time-varying shear strain, $\gamma$ (t):

$$\gamma \text{ (t)} = \gamma_0 \sin (\omega t) \tag{1}$$

with amplitude $\gamma_0$ and angular frequency $\omega$. As materials exhibited viscoelastic behaviour, the response of materials was obtained as a phase-shifted ($\delta$) shear stress function, $\tau$ (t):

$$\tau \text{ (t)} = \tau_0 \sin (\omega t + \delta) \tag{2}$$

where $\tau_0$ represents the shear stress amplitude. We can further expand the expression for shear stress as:

$$\tau \text{ (t)} = \tau_0 \sin (\omega t) \cos\delta + \tau_0 \cos (\omega t) \sin\delta \tag{3}$$

As it is shown in Equation (3) the stress consists of a component that is in phase with the strain and one being out of phase with the strain. The stress–strain relationship can thus be defined as storage modulus, G′ that is in phase with shear strain and loss modulus, G″ that is out of phase with strain:

$$G' = \tau_0/\gamma_0 \cos\delta \tag{4}$$

$$G'' = \tau_0/\gamma_0 \sin\delta \tag{5}$$

The storage modulus, G′ represents the elastic behaviour of a material, while the loss modulus, G″ represents the viscous behaviour of a material.

For each material, several repetitions of the same measurements were performed. Strain amplitude dynamic tests were performed at a constant frequency of 6.28 rad/s and by increasing strain amplitude from 0.01 to 100%. Frequency sweep dynamic tests were performed in frequency range 600 to 0.03 rad/s at the constant strain in the linear viscoelastic range, which depended on the sample and was for each sample previously determined by strain amplitude dynamic tests.

### 2.4. Chemical Analysis of Binder and Fillers

The infrared absorption spectra of the binder components, fillers, and composites were measured by attenuated total reflection spectroscopy (ATR) using a Perkin Elmer Spectrum GX NIR FT-Raman spectrometer (Perkin Elmer, Boston, MA, USA). The spectrum of the samples was measured in the range of 400 to 4000 cm$^{-1}$ at room temperature. For each spectrum, the average of 32 measurements was calculated.

The chemical surface analysis of the filler particles (steel, titanium alloy, aluminium alloy, and glass) was performed by X-ray photoelectron spectroscopy (XPS) (K-Alpha X-ray Photoelectron Spectrometer, Thermo Fisher Scientific, Erlangen, Germany). The X-ray source type employed monochromated Al K$\alpha$ radiation (1486.6 eV). The survey scan was carried out with a pass energy of 200 eV and an energy resolution of 1.0 eV. The narrow resolution spectra were recorded with a pass energy of 10 eV and step size (resolution) of 0.1 eV steps. The peaks were fitted using a Gaussian/Lorentzian mixed-function employing Shirley background correction (Data Analysis Software—Thermo Avantage v5.906, Thermo Scientific (Waltham, MA, USA)). All analyses were performed at room temperature. Hydrogen was not recorded in XPS measurements and was therefore omitted in the calculation of the elemental composition

## 3. Results

### 3.1. Strain Amplitude Sweep

The influence of filler loading on storage, G′ and loss, G″ modulus as a function of shear strain for each material is presented in Figure 3.

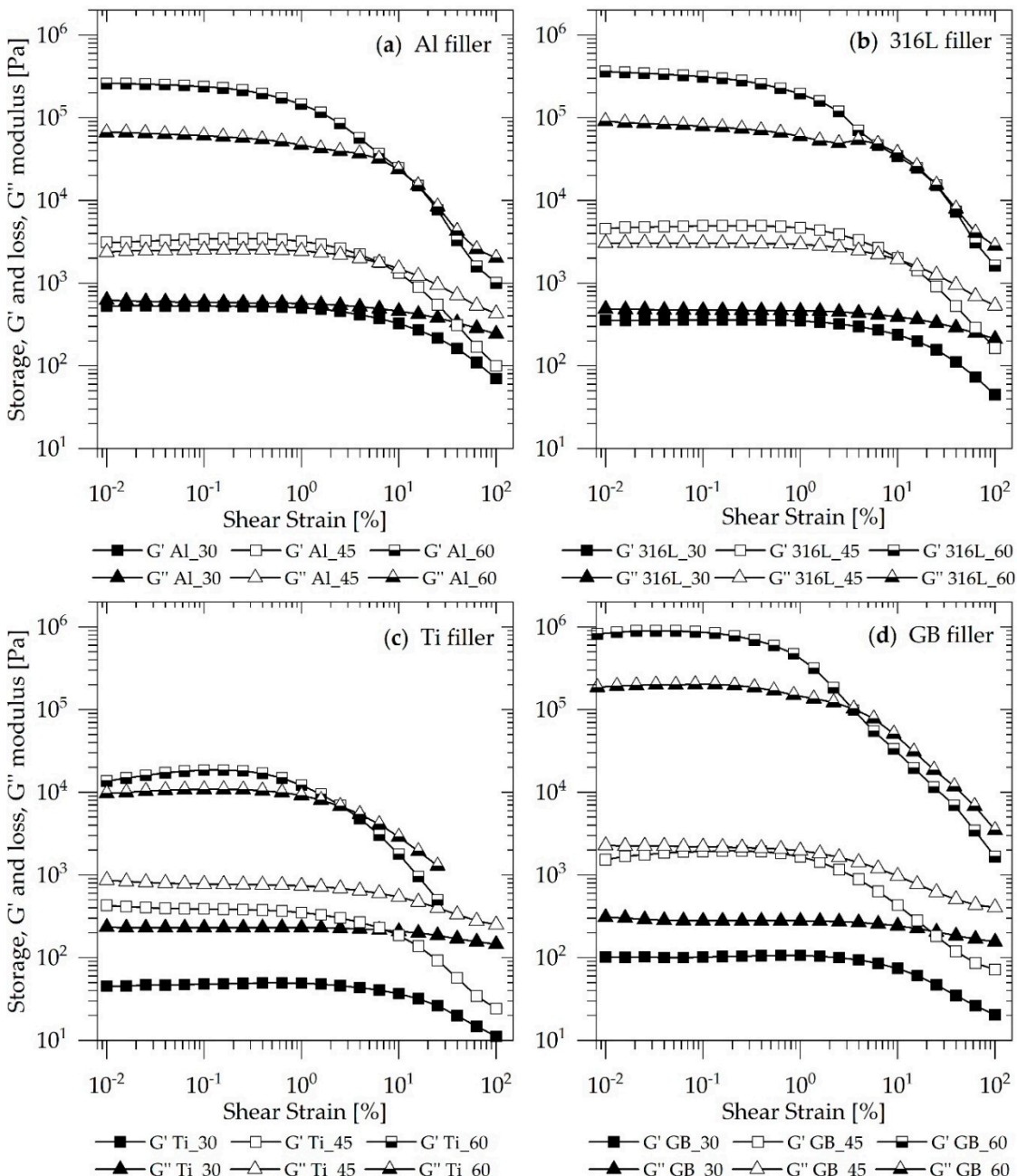

**Figure 3.** The effect of filler concentration on storage, G′ and loss, G″ modulus as a function of shear strain for the fillers: (**a**) Al, (**b**) 316L, (**c**) Ti, and (**d**) GB at 175 °C.

Figure 3 shows a well-known fact that by increasing filler concentration the storage, G′ and loss, G″ moduli are increasing [21,34,37,38]. The addition of rigid filler particles increases the overall elasticity of the composite material, hence the storage modulus G′ increases. On the other hand, the loss modulus, G″ is increased due to higher dissipation in the composite material, that occurs because of increased friction between the particles [21].

It can also be seen from the results that with higher filler concentration, the behaviour of the materials changes. Figure 4 presents the values of G′, G″ and phase shift, δ as a function of filler content. All parameters were obtained at constant frequency ω = 6.28 rad/s and small strains (γ = 0.05%), which correspond to the range of linear viscoelastic response and the conditions, where the particle–particle interactions dominate the material behaviour.

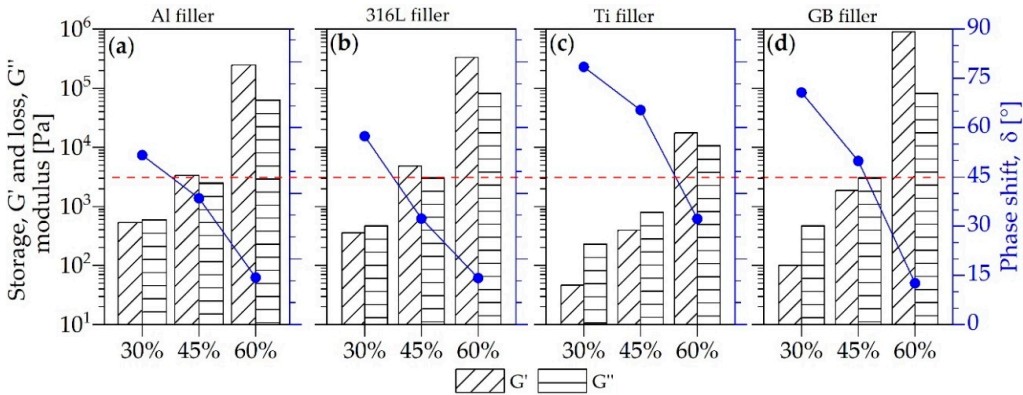

**Figure 4.** Storage, G′, loss, G″ modulus, and phase shift, δ at constant γ = 0.05%, 175 °C, and at 6.28 rad/s for the fillers: (**a**) Al, (**b**) 316L, (**c**) Ti, and (**d**) GB.

The results showed that for each particle loadings, the values of dynamic moduli were similar, or in the same range of magnitude, indicating the viscoelastic behaviour of the composites. The storage and loss modulus increased with increasing filler concentration. Moreover, the transition of the material behaviour from liquid-like (G″ > G′) to solid-like (G″ < G′) with increasing filler concentration could be observed for all the composites. The red dotted line in Figure 4 displays the δ = 45° value (G′ = G″) indicating the liquid to solid transition. At 30 vol.% of filler loading, all materials exhibited values of δ > 45°, indicating the viscoelastic liquid-like behaviour. In the case of Al and 316L materials, δ values dropped below 45° at concentrations 45 and 60 vol.%, indicating the viscoelastic solid-like behaviour. Finally, in the case of Ti and GB materials, the transition to solid-like behaviour occurred only at the highest concentrations, i.e., 60 vol.%. It should be emphasised that even at the highest concentrations (i.e., solid-like behaviour), the viscous contribution to viscoelastic behaviour was significant, suggesting the good processability of highly filled composites, studied. Results suggest that since we are comparing the composites with comparable size distributions of filler particles and binder composition, the interactions particle–binder, and particle–particle interactions (surface chemistry) play a crucial role in determining the rheological behaviour of the composites.

Strain amplitude tests were primarily performed to determine the range of linear viscoelastic response (LVR). Figure 5 presents the influence of filler material for a particular loading level on the shear strain-dependent storage modulus G′.

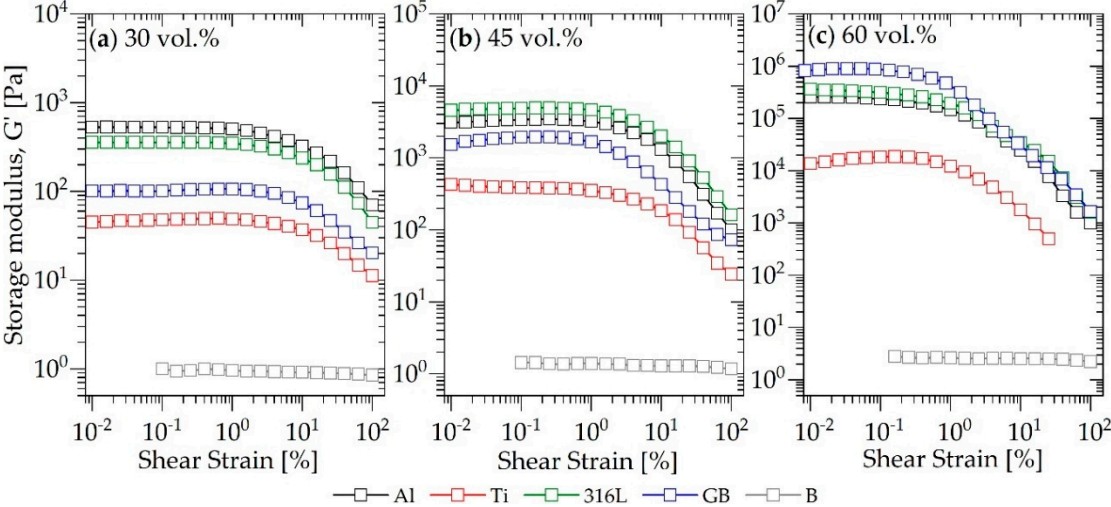

**Figure 5.** Strain amplitude sweep tests and the effect of filler material on G′ for (**a**) 30 vol.%, (**b**) 45 vol.%, (**c**) 60 vol.% filler loading at 175 °C and 6.28 rad/s.

The results showed that increasing filler concentration increased the consistency of the composites, which could be observed in the increased values of storage and loss modulus (Figure 3). However, the consistency increase was not the same for all types of filler material or filler loadings, respectively. Since all particles, used in this investigation, were spherical with comparable size distribution and embedded in the same binder system, the differences in the behaviour of the composites could be attributed either to the particle–binder or particle–particle interaction. The lowest values of the storage modulus, G′ were determined for Ti particles, irrespective of the filler concentration. This could be attributed to a weaker particle network formation or weaker particle–binder interaction. For 30 vol.% of filler loading (Figure 5a) the highest values of G′ was determined for the composite with Al particles followed by the 316L composite. The composites with these two types of particles showed the highest values of G′ also for 45 vol.% of particle loading (with a slight prevalence of the G′ for 316L), while for the highest concentration of the particles (Figure 3c), the highest values of G′ were determined for the GB composite. Results showed that the filler loading had a greater effect on GB material compared to the other three materials and that filler loading had a similar effect on 316L and Al composites. As a reference, the storage modulus of the binder systems is for all concentrations shown in Figure 5. As expected, the binder exhibited the storage modulus orders of magnitude lower compared to the other composite materials.

From Figure 5, it can also be seen that as the strain increased, the storage modulus started to decrease. This is well-known phenomena (also referred to as Payne effect) that is characteristic for filled elastomers, highly filled suspension, and filled polymers [39–48]. In general, there are two reasons for storage modulus decrease with increasing strain. The first can be attributed to polymer and chain disentanglement. Usually, disentanglements appear at larger shear strain, resulting in a broader linear plateau. The second reason for modulus decrease can be attributed to the breakdown of the particle network [34,39]. As it can be seen in Figure 5, nonlinear behaviour was not observed for the binder system in the measured range of shear strain; thus, it can be concluded that particle–particle interactions are responsible for the decrease of G′ of composite materials. Moreover, it can also be seen that the increased filler concentration intensifies the drop of the storage modulus.

### 3.2. Frequency Sweep Test

Figure 6 presents the influence of filler material on storage G′ and loss G″ modulus as a function of angular frequency for the filler loading level. Measurements were performed at constant shear strains in the range of linear viscoelastic response and constant temperature of 175 °C. For clarity, storage and loss modulus are shown separately.

It can be seen from Figure 6 that all materials exhibited frequency-dependent behaviour of the moduli. However, two distinct regions can be observed at high and low frequencies. In the range of high frequencies (above ≈100 rad/s) there is no visible difference between the response of the composite materials since the values of G′ and G″ converged or were parallel in log-log scale. This indicates that the behaviour of the composites at high frequencies was dominated by the binder system (matrix) and that the contribution of filler particles was less significant. Similar findings were reported for different highly filled composites; for example, SBR rubber and carbon-black and silica particles [44], polydimethylsiloxane (PDMS) and spherical glass beads [49], polyethylene terephthalate (PET) and ferrite nanoparticles [50], and PP and ceramic powder [51]. In the low-frequency region (below ≈10 rad/s) the flattening of the G′ curve and to a lesser extent also G″ curve could be observed. The flattening of the curve corresponds to frequency-independent behaviour and indicates that the particle–particle interactions become dominant at lower frequencies [21,24,52]. Frequency independent behaviour of filled composites has also been observed for different fillers and binder combinations [40,44,49–51,53].

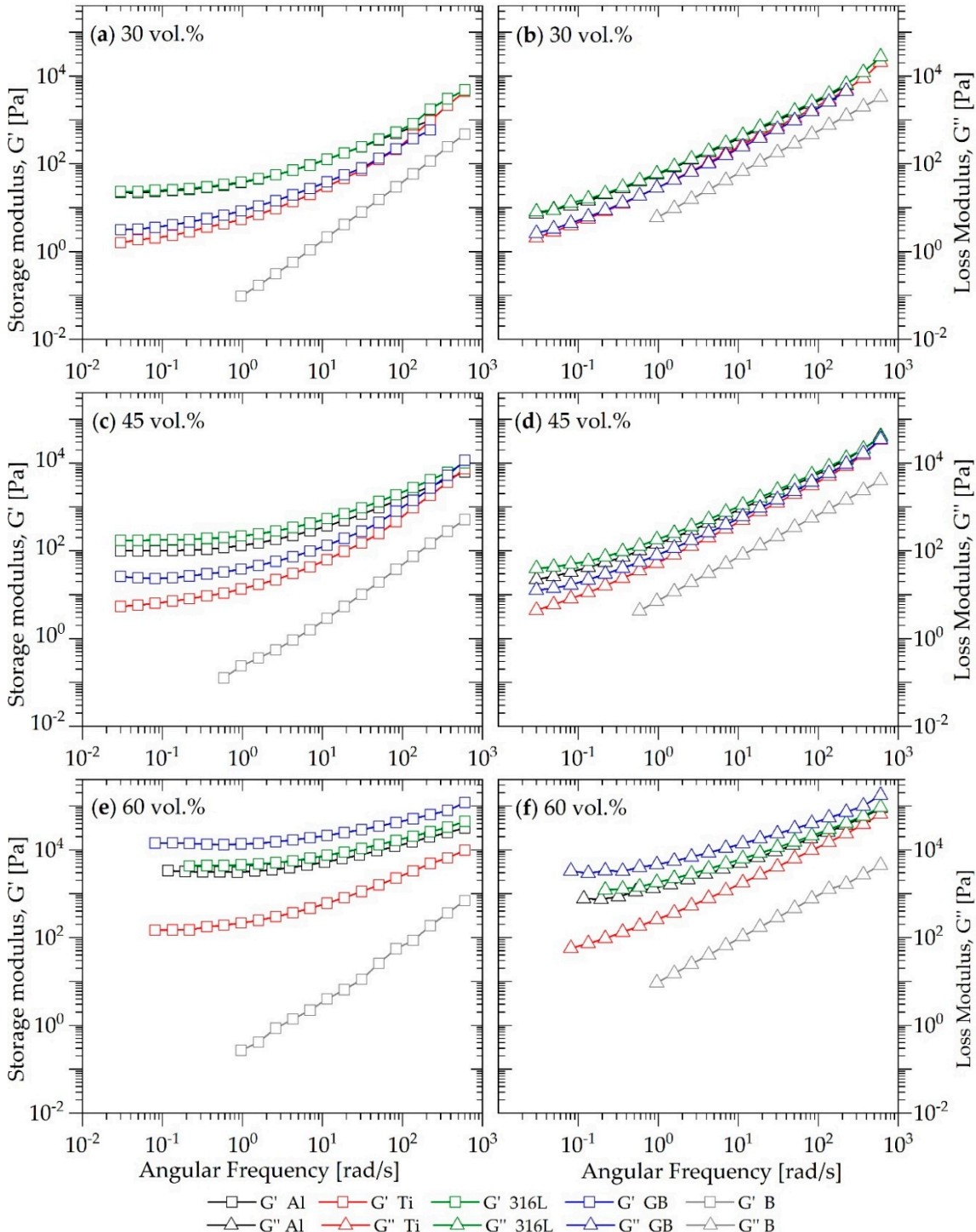

**Figure 6.** The influence of filler material on storage G′ and loss G″ modulus in the frequency domain for all filler loadings: (**a**) G′ for 30 vol.%, (**b**) G″ for 30 vol.%, (**c**) G′ for 45 vol.%, (**d**) G″ for 30 vol.%, (**e**) G′ for 60 vol.%, and (**f**) G″ for 60 vol.%.

The comparison of the frequency behaviour of composite materials at different loading levels (Figure 6) revealed that, as in the case of amplitude sweep tests, the viscoelastic behaviour in LVR depends on the type of filler material. However, in the case of 30 vol.% filler loading, there was no difference between dynamic behaviour (G′ and G″) of Al and 316L material, respectively, in the whole frequency range studied. As can be seen in Figure 6a,b, the curves for those two fillers overlap entirely

over the whole frequency range. Further, the composites with GB and Ti particles at 30 vol.% exhibited lower values of G′ and G″ compared to Al and 316L. The differences became larger at low frequencies indicating weaker particle–particle interactions compared to Al and 316L particles. Moreover, at lower frequencies, the composites with GB particles exhibited slightly higher values of G′ compared to the composites with Ti particles, which would suggest that at longer times stronger network was formed for GB material at this concentration.

The behaviour of the composites at 45 vol.% loading, shown in Figure 6c,d, was at high frequencies similar to the behaviour of the composites at the lowest concentration, which suggests that in the range of short times the binder system governs the material behaviour. The results of frequency tests at low frequencies suggest that in the composites with 316L particles stronger network was formed compared to the composites with Al particles. At the highest loading level (Figure 6e,f), we observed that the composites with GB, Al, and 316L exhibited very weak frequency dependence. The G′ curve was independent on the frequency in a wide frequency range, representing strong solid-like behaviour. On the other hand, at this loading, the composite with Ti particles showed frequency-dependent behaviour in a wide frequency range, and the G′ curve started to flatten at around 1 rad/s. It can also be seen that the values of storage modulus for the composite with GB particles at 60 vol.% were higher compared to all other materials, while the composites with 316L and Al exhibited remarkably similar behaviour. This indicates the strongest particle network formation for GB particles and the weakest for Ti particles at this concentration.

The influence of filler concentration on dynamic properties in the LVR is shown in Figure 7 for all four filler particles separately.

As expected, the increasing filler concentration increased storage, G′ and loss, G″ modulus, respectively, for all the composites used (similar as in amplitude sweep tests). As discussed, the behaviour of Al and 316L materials were similar and can also be viewed in a side-by-side comparison in Figure 7a,b. In both cases, the increase of filler content from 30 to 45 vol.% resulted in an increase of G′ for approximately one order of magnitude in the whole frequency range. Much higher increase can be observed for 60 vol.%, where G′ drastically increased, for example at frequency 10 rad/s from 130 Pa to around 5000 Pa.

On the contrary, a much smaller increase of the viscoelastic properties could be observed for Ti particles (Figure 7c), as the values of G′ from 30 to 60 vol.% increased by ≈200 Pa in the low-frequency area. The behaviour of GB material was at 30 and 45 vol.% similar to the behaviour of Ti particles, however at 60 vol.% this composite showed the highest change of properties and the values of G′ and G″ were even higher than those obtained for Al and 316L composites.

There are several ways to characterise the transition from liquid to solid-like behaviour [37,54,55]. For example, the transition can be determined when the slope of G′ curve changes to 0, when tan δ is frequency independent or at the crossover of G′ and G″ functions (G′ = G″). From Figure 7 it can be seen that the crossover point shifted to higher frequencies as the filler loading increased in all composite materials studied. The crossover point was separately analysed and is presented in Figure 8.

The transition from solid to liquid-like behaviour for Al and 316L materials at the lowest concentration occurred at low frequencies, below 1 rad/s, while the frequency of the transition for 45 vol.% increased to around 1 rad/s. For the composites with 60 vol.% of Al and 316L particles, respectively, the transition occurred at around 20 rad/s. The behaviour of Ti materials, at all concentrations, was liquid-like in the broadest frequency range. The transition at 60 vol.% occurred at 0.6 rad/s and for concentrations even lower. The crossover point at 30 vol.% was extrapolated from the experimental results since the limit of the lowest measuring frequency was above the transition. In the case of GB filler, the behaviour at concentrations 30 and 45 vol.% loosely followed the behaviour of Ti material, with crossover points at around or below 0.1 rad/s. However, at the highest concentration, the crossover point occurred at the highest frequency among all four materials, i.e., at 108 rad/s.

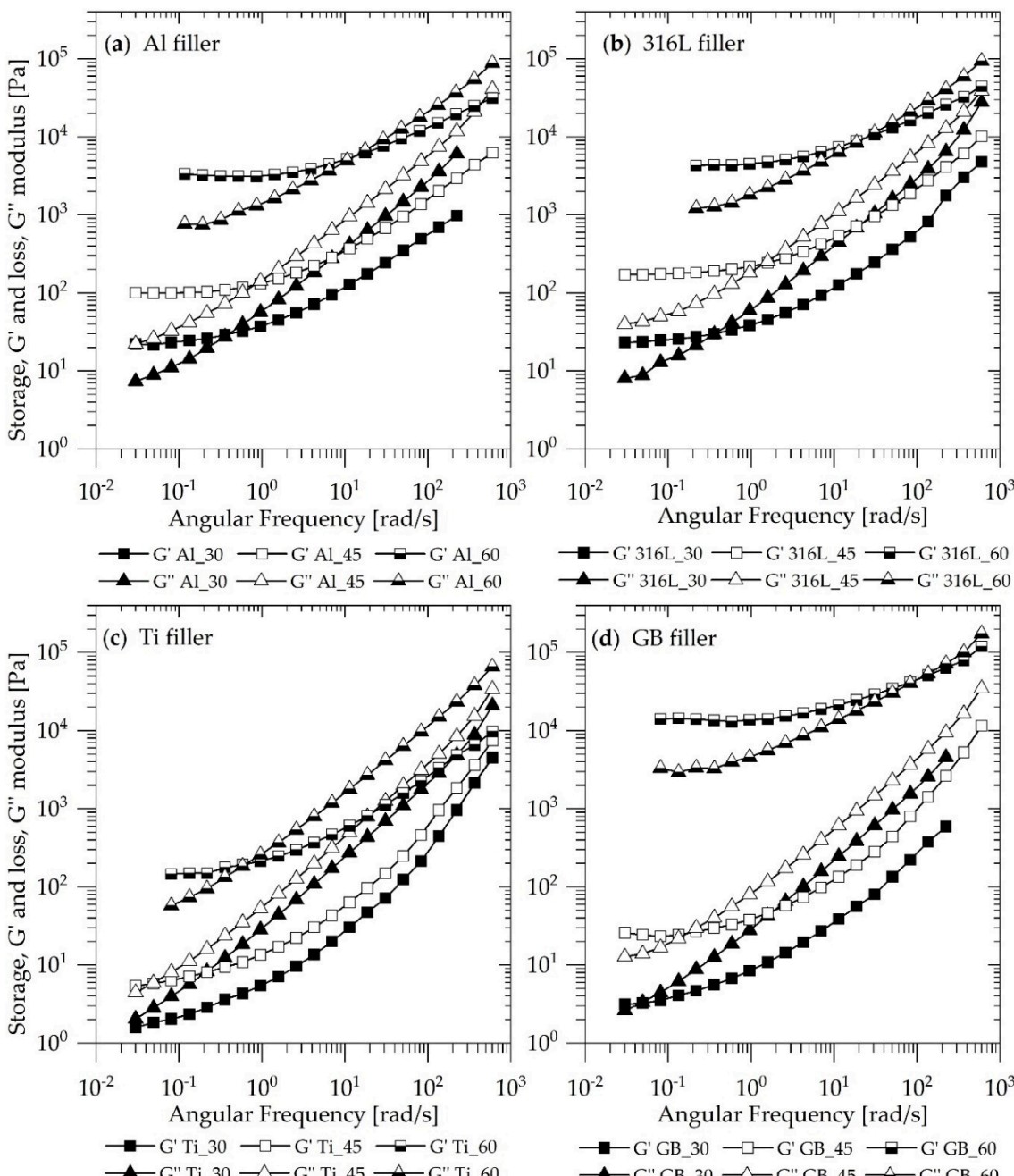

**Figure 7.** The effect of filler concentration on storage, G′ and loss G″ modulus as a function of angular frequency: (**a**) Al filler, (**b**) 316L filler, (**c**) Ti filler, and (**d**) GB filler.

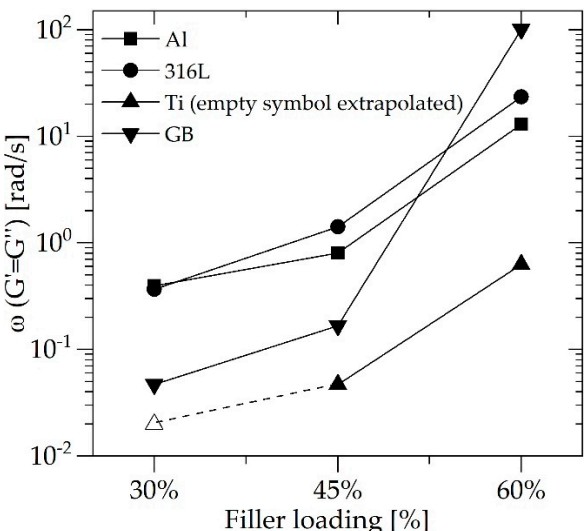

**Figure 8.** Crossover point ω (G′ = G″) for the composites with various loadings. The point at 30 vol.% for Ti was extrapolated from the measured results.

### 3.3. X-ray Photoelectron Spectroscopy

The chemical composition of the surface of each filler was measured with XPS. The survey spectra with characteristic peaks for each filler is shown in Figure 9, and elemental composition for each material is given in Table 5. As can be seen, all materials exhibited a significant amount of oxygen on the surface. Therefore, the elemental spectra were peak-fitted to define the corresponding oxidation states of different metals and metalloid Si.

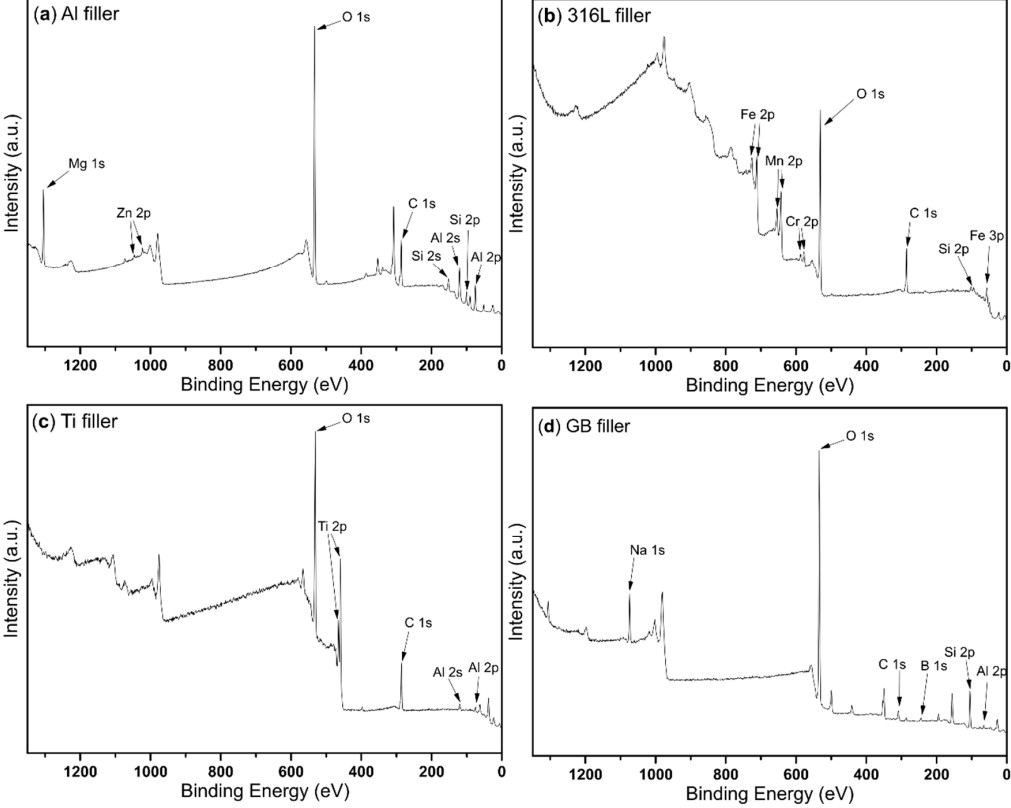

**Figure 9.** XPS survey spectra with characteristic peaks for the fillers: (**a**) Al, (**b**) 316L, (**c**) Ti, and (**d**) GB.

**Table 5.** Elemental composition of the surface of the fillers measured by XPS.

| Material | Element | Atomic % | Material | Element | Atomic % |
|----------|---------|----------|----------|---------|----------|
| AlSi10Mg | Oxygen | 43.24 | 316L steel | Oxygen | 47.74 |
|  | Carbon | 19.99 |  | Carbon | 29.24 |
|  | Aluminium | 19.96 |  | Iron | 13.01 |
|  | Silicon | 5.88 |  | Manganese | 7.81 |
|  | Magnesium | 10.57 |  | Silicon | 4.24 |
|  | Zinc | 0.34 |  | Chromium | 2.19 |
| Ti6Al4V | Oxygen | 47.84 | Glass | Oxygen | 55.97 |
|  | Carbon | 29.86 |  | Silicon | 26.08 |
|  | Titanium | 15.79 |  | Boron | 8.78 |
|  | Aluminium | 6.68 |  | Sodium | 3.62 |
|  | Vanadium | <0.5 |  | Carbon | 3.47 |
|  |  |  |  | Aluminium | 2.08 |

On the AlSi10Mg surface, we observed a high amount of $Al_2O_3$ at 74.6 eV in comparison to the elemental Al (0) at 73.1 eV in an atomic ratio of 7.4:1, which indicates high oxidation of the surface [56–58]. Moreover, magnesium and zinc were found in oxide forms of MgO at 1304.5 eV and ZnO at 1022.6 eV, while silicon was mainly found in the elemental state at 99.1 eV. Due to high oxidation of alloy surface, smaller silicon content of the element, mainly in the non-oxide form, was observed.

Similarly, high oxidation of the filler surface was detected for 316L stainless steel alloy. Similar findings were also reported by other authors [59,60]. Most of the iron was in a $Fe_2O_3$ form with a binding energy of 711.2 eV. The same was applied for other elements which were also in the oxide forms as $MnO_2$ at 642.0 eV, $Cr_2O_3$ at 576.6 eV, and SiO at 101.5 eV, respectively [61].

On the titanium alloy surface, we found titanium and aluminium in the form of $TiO_2$ at 458.4 eV and $Al_2O_3$ at 74.0 eV [62]. Moreover, also, this surface showed a high concentration of oxidised metals on the surface, and therefore, almost no response was observed in the spectrum.

Borosilicate glass, on the other hand, is a chemical-resistant material. The higher the $B_2O_3$ content in the glass, the more chemically resistant the material becomes [63]. From the XPS composition of boron and silicon, we estimated 55.8% of $SiO_2$ and 28.3% of $B_2O_3$ on the surface. With that high amount of $B_2O_3$ content in the surface composition, it was reasonable to assume that the surface was chemically stable and did not interact with the matrix material.

Oxides on the metal alloy surface can interact with the components of the binder system, particularly with the maleic anhydride (MA). The interaction between the filler particles and the matrix was investigated using ATR-FTIR analysis and is discussed in the following section.

### 3.4. ATR-FTIR Analysis

Figure 10 shows the ATR-FTIR spectra of a reference filler material Al alloy, Al composite materials, with all three concentrations, and binder system (functionalized PP; MAPP) in the range of 4000 and 500 $cm^{-1}$. The binder systems include MA grafted PP (MAPP), thermal stabiliser, and PP wax. The spectrum of the binder system showed characteristic peaks for polypropylene at 2950, 2917, 2868, 2842, 1456, 1374, and around 1018 $cm^{-1}$. Characteristic vibration at 2917 $cm^{-1}$ corresponds to $CH_2$ asymmetric stretching, the band at 2950 $cm^{-1}$ to $CH_3$ stretching, 2868 $cm^{-1}$ $CH_2$ asymmetric stretching. The characteristic peaks in the region at 1456 and 1374 $cm^{-1}$ were determined for $CH_3$ $CH_2$ rocking vibration. These peaks are in line with characteristic vibrations (symmetrical and asymmetrical) of hydrocarbon chains of PP, namely of $CH_3$ and $CH_2$ groups [64]. The spectrum of MAPP was investigated at wavelengths between 2000 and 1400 $cm^{-1}$. Three functional groups were present in MAPP. At 1778 $cm^{-1}$ a peak was associated with symmetric C = O stretching in cyclic anhydride. The characteristic peak at 1736 $cm^{-1}$ corresponded to the vibrations of ester groups (-C-O = O). In the 1240 $cm^{-1}$ region a characteristic vibration for C-O bond occurred. All three peaks suggest the functionalisation of PP with an anhydride, MAPP (maleic anhydride) [65].

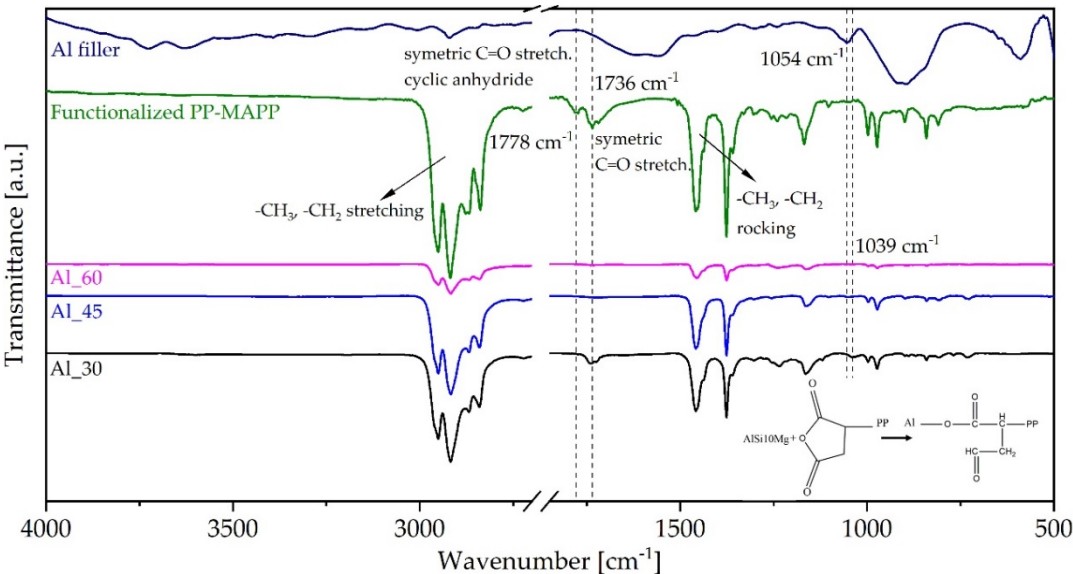

**Figure 10.** ATR-FTIR spectra of Al materials, reference binder system and Al filler material.

The aluminium materials usually show vibration peaks from 800 to 200 cm$^{-1}$. The ATR spectrum of Al filler (AlSi10Mg) indicated an oxidation process on the surface. The peak at 1073 cm$^{-1}$, can be associated with Al-O vibrations, the peak at 900 cm$^{-1}$ can be related to OH vibrations and peak at 588 cm$^{-1}$ to Al-O vibrations.

ATR spectra of Al composite materials (Al_30, Al_45, and Al_60) showed the characteristic peaks for PP, with the majority belonging to the $-CH_2$, $-CH_3$ groups of the aliphatic chain of the composites. In further analysis, we focused on the characteristic peaks for PP functionalised with maleic acid, between 1400 and 1800 cm$^{-1}$ wavelengths. It was observed that the vibration peak at 1778 cm$^{-1}$, related to cyclic anhydride PP disappeared and the vibration peak of MAPP at 1736 cm$^{-1}$ decreased. It is reasonable to assume that the MA anhydride PP ring opened and reacted with the Al filler. Additionally, a peak shift from 1073 to 1040 cm$^{-1}$ (associated with Al-O-H vibrations of filler) was observed, which indicates the binding of Al-O surface group located on Al filler material to the binder system and formation of a covalent bond between Al filler and MAPP. The same changes were observed regardless of the filler concentration. The main mechanism of chemical bonding between Al alloy and functionalised MAPP is shown in the lower right corner of Figure 10.

Figure 11 shows the characteristic vibrations of the binder system (functionalised PP-MAPP), 316L composite materials at all three concentrations and as a reference, the spectrum of 316L filler material. As expected, typical vibrations peaks of PP, MAPP, and thermal stabiliser, were detected. The 316L filler material mostly consisted of Fe alloys where vibration peaks of metallic materials usually appear in the 800 to 200 cm$^{-1}$ range. By measuring ATR spectra of the 316L filler alone (magenta curve), we identified a vibration peak at 779 cm$^{-1}$ related to Fe-vibrations.

ATR spectra of 316L composite materials (316L_30, 316L_45, and 316L_60) showed the characteristic peaks for PP and Fe characteristic vibration. Vibration peaks characteristic for the $-CH_2$, $-CH_3$ groups of the aliphatic chain of the PP were observed. In the range identified as characteristic for Fe-vibrations, a shift from 779 cm$^{-1}$ to the lower wavelengths of 729 cm$^{-1}$ for all three composite materials was observed. Similarly, as in the case of Al material, the characteristic peak of cyclic MAPP at 1778 cm$^{-1}$ disappeared (again indicating the opening of the anhydride ring), and the peak at 1736 cm$^{-1}$ associated with the vibration of C = O−O− symmetric stretching of MAPP implied the bonding of Fe- group to the binder system (to the MAPP).

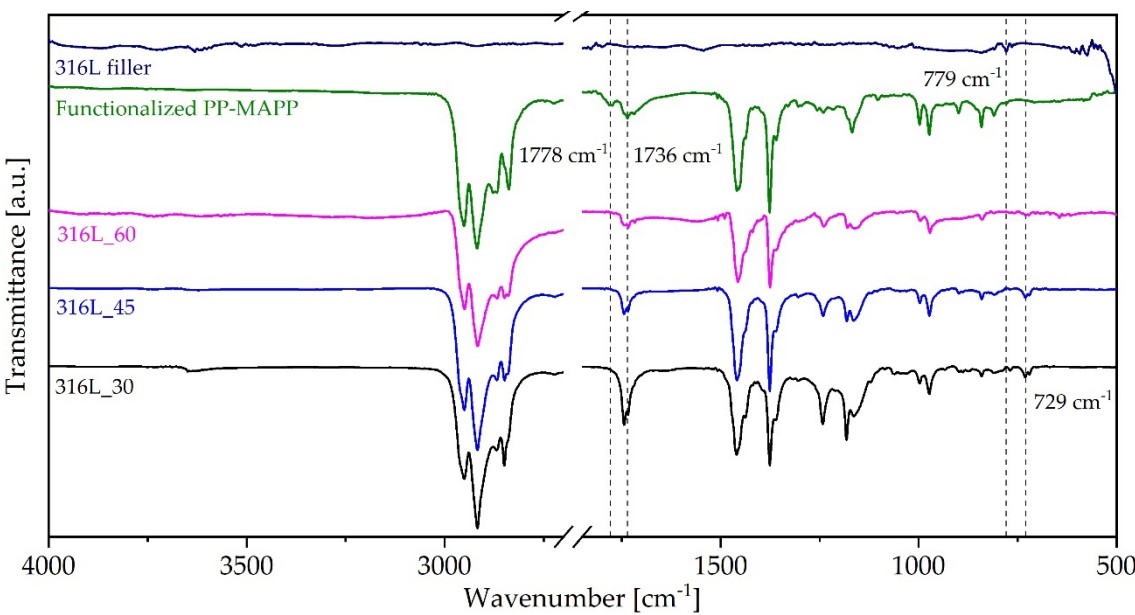

**Figure 11.** ATR-FTIR spectra of 316L materials, reference binder system and 316L filler material.

The spectra of Ti composite materials at all three concentrations, as well the binder system and just the Ti filler reference sample are shown in Figure 12. Typical vibrations of PP, MAAP, and thermal stabiliser, with the same characteristic peaks as described before were detected in Ti composite materials. The spectrum of Ti filler reference material showed characteristic peaks in the wavelength range from 800 to 600 cm$^{-1}$, and typical peaks for metal oxides was detected at 858 and 775 cm$^{-1}$ (the vibration of Ti-O bond).

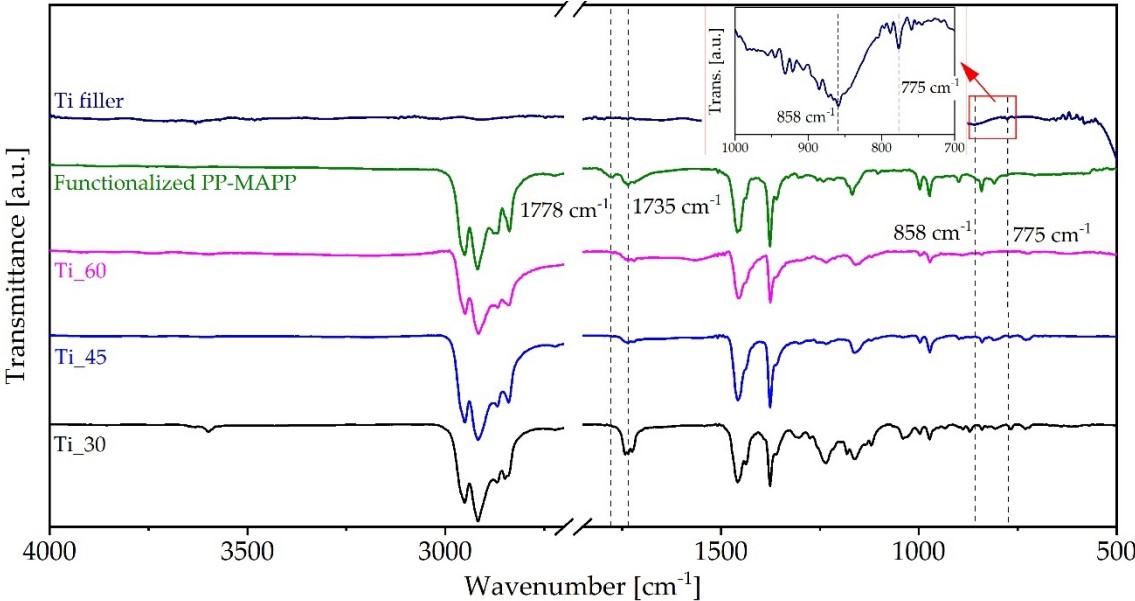

**Figure 12.** ATR-FTIR spectra of Ti materials, reference binder system, and Ti filler material.

The ATR spectra of Ti composite materials showed the characteristic peaks for PP (as in the previous two materials). Additionally, here, with the addition of filler particles, the characteristic peak of MAPP at 1778 cm$^{-1}$ disappeared indicating the opening of the MA anhydride ring. As opposed to the previous two materials, it seems that no vibration peak shifts could be detected. Chemical bonding of Ti filler material to the binder system could not be confirmed in this case. This might be due to

the limitations of this particular technique. Although, the peak at 1736 cm$^{-1}$ indicates a physical interaction between Ti-O groups and carbonyl groups of an opened ring of MAPP.

The final spectra in Figure 13 show spectra of GB composite materials at all three concentrations, a spectrum of the binder system and as a reference also the spectrum of GB filler. The spectrum of filler material (GB) showed peak vibrations at around 1040 cm$^{-1}$, typical for Si-O vibrations of glass (silicate) materials.

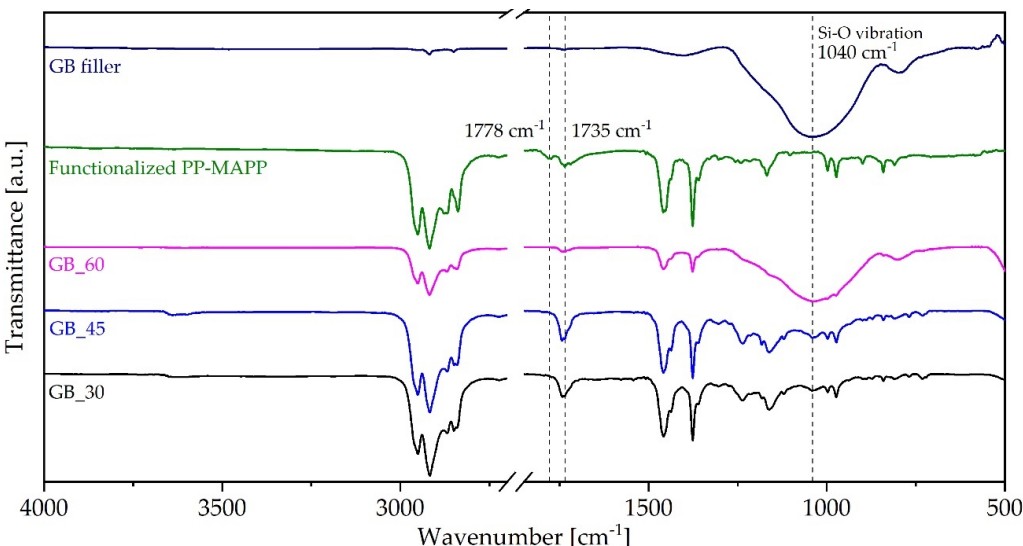

**Figure 13.** ATR-FTIR spectra of GB materials, reference binder system, and GB filler material.

In the composite materials, we detected the presence of silicates, since vibration peaks occurred at 1040 cm$^{-1}$. In this case, it was clear that there was no peak shift related to GB filler (at 1040 cm$^{-1}$) and that there was no chemical bonding between the filler material and polymeric binder system. However, as in all investigated materials, also, in this case, the peak at 1778 cm$^{-1}$ disappeared as the filler was added indicating the ring opening behaviour of MAPP. At the same time, the existing peak at 1735 cm$^{-1}$ indicated a physical interaction between Si-O groups and of carbonyl groups of an opened ring of MAPP.

## 4. Discussion

Within the presented research, we compared the rheological properties of different spherical filler materials with similar particle size distribution at three filler loadings (30, 45, and 60 vol.%). The composition of the three-component binder was the same for each filler loading level.

The results of the influence of filler loading on rheological properties showed that the increasing filler loading increased storage, G′ and loss, G″ modulus of all materials. The increase of both moduli was observed in the whole frequency range measured (frequency sweep test, Figure 7), as well as at all tested shear strains (amplitude sweep tests, Figure 3). This observation is in line with the literature [21,34,37,38]. As opposed to the general behaviour, the extent of moduli change depends on the material.

Increasing filler loading also changed the rheological character of the material. As the filler loading was increased to 60 vol.%, all four composites exhibited solid-like behaviour at a constant frequency and small shear strains (Figure 4). Moreover, filler loading affected the solid-like to liquid-like transition in the broader frequency domain (Figures 7 and 8). In general, it was observed that in the range of lower frequencies, where particle–particle interactions are dominant, all composites exhibited solid-like behaviour. As the frequency increased, the material behaviour changed to liquid-like. Furthermore, when the concentration of filler particles increased, the transition from solid-like to liquid-like behaviour

appeared at higher frequencies since the particle network was denser. The value of the frequency, where the transition occurred, depended on the filler material.

Comparison of the effect of filler materials on rheological properties showed that the composites with Al fillers and 316L filler exhibited remarkably similar rheological behaviour. This can be seen through the amplitude sweep tests (Figure 5) and frequency tests (Figure 6), where the composites with both materials exhibited similar values of storage, G′ and loss, G″ modulus regardless of the filler loading. On the other hand, it was shown that the composite materials with Ti fillers exhibited the lowest dynamic moduli in amplitude and frequency tests; again, regardless of the filler loading. The rheological behaviour of the composites with GB filler was similar to the behaviour of Ti composites at concentrations 30 and 45 vol.%, especially in the frequency domain (Figure 6). At both concentrations, the values of G′ and G″ in the low-frequency range, where particle–particle behaviour is dominant, were lower compared to Al and 316L materials. However, at 60 vol.%, the behaviour of GB composite significantly changed. G′ and G″ increased significantly compared to 316L and Al materials. To illustrate the extent of the G′ increase in the area of particle-particle dominated behaviour, the values of G′ at 30 vol.% loading and frequency ≈0.1 rad/s were compared to the values G′ at 60 vol.% (again at ≈0.1 rad/s). The storage modulus of GB composite material increased about 3500 times, while the increase of Al and 316L composite materials was between 140 and 180 times. The G′ values of Ti composite material increased the least, about 60 times.

The different behaviour of composite materials can be partially explained with the ATR-FTIR results and the occurrence of physical or chemical bonding. On the one hand, it was shown that GB composite materials interact only physically with the binder system. FTIR results indicate the adsorption of Si-O surface groups with the open ring of the carbonyl group of MAPP component of the binder system. Similarly, FTIR results for Ti composite suggest physical interaction between the filler and MAPP component of the binder system. The Ti-O surface groups reacted with the carbonyl groups of the MAPP component. No vibration peak shifts between the binder and filler were observed for Ti composite material. Thus, no chemical bonding of Ti filler material with the binder system could be clearly confirmed.

On the other hand, Al and 316L composite materials showed the adsorption of Al and Fe surface groups to the MAPP component indicating chemical bond formation. In addition, for both fillers (at all concentrations) a vibration peak shifts of Al, 316L, and carbonyl groups of MAPP were observed suggesting the formation of a covalent bond between the filler and the binder.

From the presented results, we can conclude that the chemical bonding between filler and binder causes stronger polymer–filler interactions which result in higher values of storage and loss modulus at low frequencies where material behaviour is governed by the particles (except for GB material at 60 vol.%). The same was also applied to amplitude tests, where materials 316L and Al were showing higher G′ values at all strains and concentrations (again apart from GB material at 60 vol.%). Moreover, the results suggest that if the shape and size of the particles are similar and the binder is the same, the chemical interaction between the binder and the filler determines the composite behaviour, regardless of filler material. This is clearly seen for the examples of 316L and Al materials which showed the similar strain and frequency behaviour since they both formed bonds with the binder.

To further investigate the unexpected behaviour of GB composite materials, SEM micrographs were obtained for GB composites at all three concentrations (Figure 14).

Figure 14a shows the GB composite material at 30 vol.% loading. We can see that the glass beads were embedded in the binder matrix, and no considerable damage of the glass particles could be observed. However, at the 45 vol.% loading, shown in Figure 14b, we can already see some broken glass beads, some of them presented by a red arrow. Furthermore, at 60 vol.%, more glass beads were broken (Figure 14c). Glass beads are fragile, and at high enough concentration, they start to break during the processing in the kneader and in the press. With the presented SEM micrographs, we could now understand that the increase of dynamic rheological properties compared to other materials was related to the broken glass particles, and as the concentration of filler increased, the number

of broken glass beads was increased, too. As reported in the literature [66], the glass pieces show much higher viscosity at the same filling grade than the undamaged spheres. Moreover, the frictional forces between irregular particles are higher and cause an increase in viscosity and the occurrence of shear-thinning behaviour.

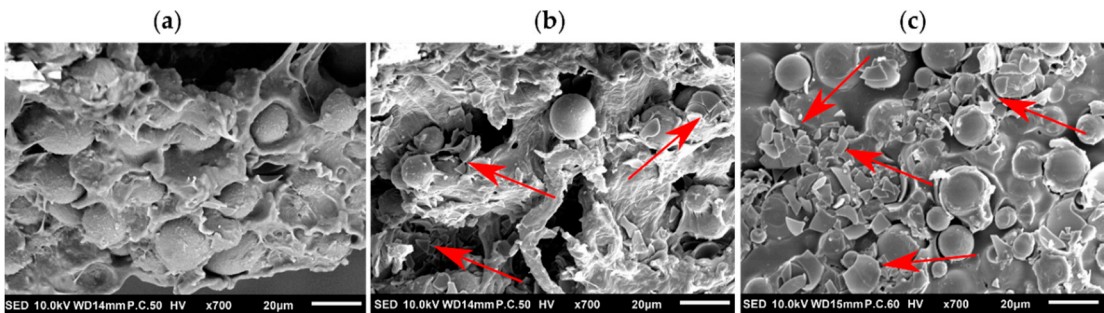

**Figure 14.** SEM micrograms of GB composite materials at 30 vol.% (**a**), 45 vol.% (**b**), and 60 vol.% (**c**).

The results of the presented research proved that the rheological behaviour of highly filled materials strongly depends on the interactions between the filler and matrix material, assuming that the same matrix material is used and that the filler material size and shape are the same. Therefore, in this particular case, it can be stated that steel and aluminium particles are forming chemical bonds between the particles and the matrix, and as such the rheological values are higher than for fillers forming physical interactions only (i.e., Ti and GB).

As the results give an insight into the particle–polymer interaction, it would be reasonable to assume that using materials that exhibit a stronger particle network and stronger particle–filler interaction would result in a more homogeneous material flow. This is of particular importance, especially in the SDS processes (Shape—Debind—Sinter), where it is crucial to obtain a homogeneous distribution of particles. The main benefit lies in the production of improved feedstocks, highly filled polymers, which will be the base for better products. The performance in the manufacturing processes of materials showing stronger filler network and the better polymer–filler connection should be further investigated with flow characterisation and trial manufacturing tests in injection moulding, extrusion, or additive manufacturing techniques.

**Author Contributions:** Conceptualisation, M.B., J.G.-G., C.K. and L.S.P.; formal analysis, M.B.; funding acquisition, L.S.P.; investigation, M.B., J.G.-G., K.P.Č., B.M.; methodology, M.B., J.G.-G., C.K., K.P.Č., B.M., L.S.P.; project administration, L.S.P., J.G.-G.; resources, M.B., J.G.-G., K.P.Č.; supervision, L.S.P.; visualisation, M.B., K.P.Č.; writing—original draft, M.B.; writing—review and editing, M.B., J.G.-G., C.K., K.P.Č., B.M., L.S.P. All authors have read and agreed to the published version of the manuscript.

**Funding:** This work was financially supported by the Slovenian Research Agency in a frame of project J2-9443 and the research core funding No. P2-0264. This Austrian–Slovenian collaboration was facilitated by the support of the Austrian Agency for International Cooperation in Education and Research (OeAD) Project No. SI 24/2020 and by the Slovenian Research Agency project no. BI-AT/20-21-028.

**Acknowledgments:** The authors acknowledge the help of Sebastjan Matkovič from Faculty of Mechanical Engineering, University of Ljubljana for making SEM images of GB composite materials. We would also like to acknowledge Matjaž Rejec, Anton Paar d.o.o., Slovenia for performing the filler materials particle size analysis.

**Conflicts of Interest:** The authors declare no conflict of interest. The funders had no role in the design of the study; in the collection, analyses, or interpretation of data; in the writing of the manuscript, or in the decision to publish the results.

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
