# Peer review of "Rheological Behaviour of Highly Filled Materials for Injection Moulding and Additive Manufacturing: Effect of Particle Material and Loading"

_applsci, doi:10.3390/app10227993_

Round 1

Reviewer 1 Report

Manuscript concerns experimental study of the rheological behaviour of highly-filled inorganic powder boded with use of thermoplastic polymers. Authors demonstrated impact of chemical interactions between the filler and matrix material and the particle shape and size on rheological behaviour of composites. Manuscript is well written. It provides some new scientific information. Manuscript is in the scope of journal. It can be recommended for publication after very little improvements.

Particular comments:

Table 4. Pressure should be expressed in SI units i.e. Pa.

Line 203. Justification of selection of temp. 175o for rheological tests should be provided.

Reviewer 2 Report

The comment is in the attached MS-Doc file

Reviewer 3 Report

The paper is well written. All experiments carried out as well as the results and the conclusions are described in detail and can be understood easily. The overall approach to investigate such material systems is consistent and well in line with the literature and the state of the art in this area.

Regarding the presentation, everything is clearly provided, the graphs are consistent and well prepared as well as explained. The conclusions drawn from the experiments are also consistent and comprehensible.

Having read the paper, a few open questions are left behind:

  • At the beginning of the discussion, the authors state that “the composition of the three-component binder was the same for each filler loading level”. While this is true for the actual components used, the fractions of them being used with different filler loadings vary (see table 1). There is no explanation why this variation has been carried out.
  • On a similar note: The fillers are described with the regards to their particle size, surface chemistry and particle shape, but no information is given about their surface area. Similar research in this area states that the surface area has a major impact on the rheological behavior of such highly filled systems.
  • Why did the others choose to stop at 60 Vol%? Literature shows that higher contents are possible (up to 70 Vol%), even with smaller particles. The impact on the rheological properties becomes even higher at such high filler loadings, so it would be interesting to investigate the behavior at the border of what is possible.
  • While the title suggest an application oriented approach, no examples are given for actual application of these materials, whether in injection molding nor additive manufacturing. That could be due to the fact that this to be presented in a follow-up paper. But as there is also no outlook given at the end, the actual applicability of the investigated materials remains unclear.

Judging the innovation brought to this area, I am somewhat divided. While I am not fully aware of the research conducted on the presented filler materials, the general approach and most of the conclusions have already been described in the literature for the same or similar materials.

Also, considering the particle sizes of the fillers used in this research it is already known that higher volume loadings can be achieved even with lower particle sizes. So it can be stated that while this paper brings additional knowledge to the area, it is not highly innovative.

Lastly, there is a small typo on page 2, line 87: “steric acid” should be “stearic acid”.

Overall, even considering the mentioned shortcomings I think the paper can be considered for publication after major revision as it is very well written, contributes to the gain of knowledge in that specific field and could be helpful to other research groups.

Reviewer 4 Report

The paper deals with the composition of thermoplastic polymer embedding inorganic fillers. It shows how these inclusions affect the rehology.

The paper is overall well written. Therefore, I recommend it for publication.

Round 2

Reviewer 2 Report

The authors have addressed all the comments and queries. 

Reviewer 3 Report

The authors have satisfactorily addressed the reviewer's comments and the manuscript has been significantly improved.